# Cranberry Proanthocyanidins as a Therapeutic Strategy to Curb Metabolic Syndrome and Fatty Liver-Associated Disorders

**DOI:** 10.3390/antiox12010090

**Published:** 2022-12-30

**Authors:** Francis Feldman, Mireille Koudoufio, Ramy El-Jalbout, Mathilde Foisy Sauvé, Lena Ahmarani, Alain Théophile Sané, Nour-El-Houda Ould-Chikh, Thierry N’Timbane, Natalie Patey, Yves Desjardins, Alain Stintzi, Schohraya Spahis, Emile Levy

**Affiliations:** 1Research Centre, Sainte-Justine University Health Center, Montreal, QC H3T 1C5, Canada; 2Department of Nutrition, Université de Montréal, Montreal, QC H3T 1J4, Canada; 3Department of Radiology, Université de Montréal, Montreal, QC H3T 1J4, Canada; 4Department of Pathology, Université de Montréal, Montreal, QC H3T 1J4, Canada; 5Institute of Nutrition and Functional Foods, Laval University, Quebec, QC G1V 4L3, Canada; 6Department of Biochemistry, Microbiology, and Immunology, Faculty of Medicine, Ottawa Institute of Systems Biology, University of Ottawa, Ottawa, ON K1H 8M5, Canada; 7Department of Biochemistry & Molecular Medicine, Université de Montréal, Montreal, QC H3T 1J4, Canada

**Keywords:** proanthocyanidins, metabolic syndrome, insulin resistance, hyperlipidemia, oxidative stress, inflammation

## Abstract

While the prevalence of metabolic syndrome (MetS) is steadily increasing worldwide, no optimal pharmacotherapy is readily available to address its multifaceted risk factors and halt its complications. This growing challenge mandates the development of other future curative directions. The purpose of the present study is to investigate the efficacy of cranberry proanthocyanidins (PACs) in improving MetS pathological conditions and liver complications; C57BL/6J mice were fed either a standard chow or a high fat/high sucrose (HFHS) diet with and without PACs (200 mg/kg), delivered by daily gavage for 12 weeks. Our results show that PACs lowered HFHS-induced obesity, insulin resistance, and hyperlipidemia. In conjunction, PACs lessened circulatory markers of oxidative stress (OxS) and inflammation. Similarly, the anti-oxidative and anti-inflammatory capacities of PACs were noted in the liver in association with improved hepatic steatosis. Inhibition of lipogenesis and stimulation of beta-oxidation could account for PACs-mediated decline of fatty liver as evidenced not only by the expression of rate-limiting enzymes but also by the status of AMPKα (the key sensor of cellular energy) and the powerful transcription factors (PPARα, PGC1α, SREBP1c, ChREBP). Likewise, treatment with PACs resulted in the downregulation of critical enzymes of liver gluconeogenesis, a process contributing to increased rates of glucose production in type 2 diabetes. Our findings demonstrate that PACs prevented obesity and improved insulin resistance likely via suppression of OxS and inflammation while diminishing hyperlipidemia and fatty liver disease, as clear evidence for their strength of fighting the cluster of MetS abnormalities.

## 1. Introduction

The metabolic syndrome (MetS) reflects a constellation of cardiometabolic risk factors that increase cardiovascular diseases (CVD) [1]. This growing scourge is partly attributed to poor nutritional habits and a sedentary lifestyle, which may lead to obesity, dyslipidemia, insulin resistance (IR) and cardiovascular disorders, such as metabolic-associated fatty liver disease (MAFLD) and type 2 diabetes [2,3,4]. The mismanagement of MetS or failure to blunt its progression represents a compelling challenge for any modern society and healthcare system, with a worldwide prevalence estimated at 30% in 2021 [5]. Not only is knowledge of the putative mechanisms necessary to curb the MetS, but it is also mandatory to uncover optimal therapeutic agents capable of targeting the set of symptoms. Since nutrition represents the first-line treatment in most cardiometabolic disorders [1], it is plausible that the inclusion of functional foods and bioactive phytonutrients (i.e., polyphenols) in the day-to-day regimen could prove a promising strategy. It should be noted that several studies have already emphasized the prophylactic effects of polyphenol-enriched fruit extracts on human metabolic disorders [6,7,8,9,10,11].

As bioactive molecules, polyphenols play a significant role in many physiological and metabolic processes, partly due to their antioxidant, anti-inflammatory, and lipid-lowering properties [12,13]. Their preventive and curative actions were outlined in various chronic illnesses, including oncologic, neurodegenerative, infectious, metabolic, and CVD [14,15,16,17]. Despite the wide popularity of their phenolic components, the mechanisms of action are not well understood. In recent years, berries have piqued interest due to their high content in polyphenolic compounds in association with health benefits [15,18,19,20,21]. These small, pulpy fruits, namely Aronia, grape, buckthorn, blue-, red- and cranberries, are rich in anthocyanins, proanthocyanidins (PACs), phenolic acids and flavonols [22]. Among the flavonoids, PACs, representing complex and colourless polymers of flavan-3-ol monomer units (e.g., catechin and epicatechin), are endowed with broad health promoting activities while conferring flavor and displaying nutritional values [23,24,25]. Additionally, health-relevant attributes described for berries were ascribed to polyphenols despite their content enriched in other nutritive compounds such as polysaccharides, vitamins, fatty acids, amino acids, nucleosides, essential minerals, and dietary fibers [20,26]. Moreover, differences in the beneficial influence of diverse berry species depend on the composition of their PACs content due to their distribution areas and climate.

Cranberries are unique among fruits as they have high content of polyphenolic compounds with a wide range of biological effects [27,28]. Our ability in isolating small, medium, and high molecular mass polyphenols from cranberries allowed us to freshly compare their anti-oxidative and anti-inflammatory properties, and their potential to regulate mitochondria dysfunction and oxidative stress (OxS) in intestinal Caco-2/15 cells [29]. Recently, we succeeded in purifying the cranberry-PAC fraction to highlight the mechanisms modulating insulin sensitivity and metabolic pathways in intestinal Caco-2/15 cells [30]. Despite the relative simplicity and experimental control of in vitro models, the importance of evaluating the physiological role of PACs in vivo is essential for capturing the inherent complexity of organ systems and ultimately facilitating translatability to humans. Therefore, the present study explores the potential amelioration of cardiometabolic risk factors by PACs in C57BL/6 mice, maintained on an obesogenic high fat/high sucrose (HFHS) diet.

## 2. Materials and Methods

### 2.1. Purification and Characterization of PAC Polymeric Fraction

Cranberry powdered extract was obtained from Diana Food Canada (Champlain, QC, Canada) [18]. The composition of the extract is described in Appendix A.

### 2.2. Animals

All studies and experimental procedures were conducted according to the guidelines of the animal care committee of CHU Sainte-Justine (Montreal, QC, Canada). Eight-week-old C57BL/6J male mice (Charles River; Montreal, QC, Canada) were housed in a controlled environment (23 °C; 12/12 h light-dark) with free access to food and drinking water. Following one week of acclimation on a standard diet, mice were fed either a chow (Teklad, 2018, Envigo) or HFHS (D17032403, Research Diets). Three groups were formed: Chow, HFHS and HFHS + PACs (daily doses: 200 mg/kg body weight). Of note, Chow and HFHS-nourished mice received the vehicle (water) by gavage to comply with HFHS-fed mice treated with PACs. Body weight gain and food intake were assessed twice a week. After 12 weeks on dietary regimens, animals were fasted overnight, anesthetised with isoflurane and euthanized by cardiac puncture. Blood was drawn in EDTA-treated tubes and immediately centrifuged (3000× *g* for 20 min at 4 °C) to separate plasma from cells. Adipose tissues (from perirenal, epididymal, inguinal and mesenteric regions), intestine and liver were carefully collected and weighted, then immediately flash-frozen in liquid nitrogen and finally stored at −80 °C until further analysis. Importantly, there is a real distinction between measurements of adipose tissue and other organs such as the liver and intestine. Each adipose tissue region was separately collected and weighed, allowing the distribution of the overall adipose tissue to be established. Subsequently, all the weights of the individual fat tissues were added together to determine the total amount of adipose tissue. Importantly, the liver and gut were treated separately.

### 2.3. Glucose Homoeostasis

At week 8, animals were fasted for 6 h and then an insulin tolerance test (ITT) was performed after an intraperitoneal injection of insulin (0.75 UI/kg body weight). Blood glucose concentrations were measured with an Accu-Check glucometer (Bayer) before (0 min) and after (5, 10, 15, 20, 25, 30 and 60 min) insulin injection. At the end of week 10, following an overnight fast, an oral glucose tolerance test (OGTT) was performed after gavage with glucose (1 g/kg body weight). Blood was collected before (0 min) and after (15, 30, 60, 90 and 120 min) glucose challenge for glycaemia determination. Additionally, blood samples (∼60 μL) were collected at each time point during OGTT for measurement of insulinemia using an ultrasensitive ELISA kit (Mercodia Ultrasensitive insulin ELISA, Sweden). The homeostatic model assessment for IR (HOMA-IR) index was then calculated using the following formula: fasting insulinemia (μUI/mL) × fasting glycemia (mM)/22.5.

### 2.4. Biochemical Analysis

Concentrations of plasma triglycerides (TG) and total cholesterol (TC) were quantified using commercial kits (Randox Laboratories, Crumlin, UK; Fujifilm Wako, Lexington, KY, USA). For lipoprotein isolation, unfrozen whole plasma samples were initially pooled, then submitted to sequential gradient ultracentrifugation for lipoprotein separation using a TLA-110 rotor in an Optima™ MAX Ultracentrifuge (Beckman Coulter Life Sciences Headquarters, Indianapolis, IN, USA). Briefly, centrifugations were operated at 4 °C for isolation of Very low-density lipoprotein (VLDL:1.006 g/mL, 90,000 rpm, 3 h) and High-density lipoprotein (HDL: 1.21 g/mL, 40,000 rpm 48 h). Following separation, lipoprotein fractions were frozen at −80 °C for further analysis. Plasma lipopolysaccharide (LPS, MyBioSource, San Diego, CA, USA), F2-isoprostane levels and malondialdehyde (MDA) concentration were assessed as described previously [31].

### 2.5. Hepatic Lipid Analysis and Histology

Liver specimens were homogenized and lipids were isolated following Folch chloroform-methanol extraction [32]. Liver triglycerides and cholesterol were measured and assessed using commercial kits (Randox laboratories). For tissue histology, the liver specimens were fixed in 10% formalin overnight and then embedded in paraffin. Sections were obtained with a microtome and immediately stained with hematoxylin-eosin, and examined by light microscopy. Images of the stained tissues were captured with a Zeiss Imager A1 For the tissue histology, the liver specimens were fixed in 10% formalin overnight and then embedded in paraffin. Sections (3 µm) were obtained with a microtome and immediately stained with hematoxylin-eosin, and examined by light microscopy. Images of the stained tissues were captured with a Zeiss Imager A1 and measurements were evaluated with the Axiovision software [31].

### 2.6. Western Blot Analysis

Liver samples (0.1 g) were first homogenized in 1 mL of Ripa cell lysis buffer containing NP-40 (0.5%). Protein concentration was determined using the Bradford method (Bio-Rad, Hercules, CA, USA) with bovine serum albumin as a standard. Following denaturation in SDS/ß-mercaptoethanol sample buffer, 15 μg-proteins were run in 10% SDS-polyacrylamide gels and then electroblotted on nitrocellulose membranes. Fat-free milk was used for the initial blocking of non-specific sites, followed by overnight incubation at 4 °C with the following primary antibodies at 1/1000 dilution unless otherwise specified: Sterol regulatory element binding protein-1c (SREBP1c), Peroxisome proliferator-activated receptor alpha (PPARα) from Cayman Chemical); Fatty acid synthase (FAS), Acetyl-CoA carboxylase (ACC), AMP-activated protein kinaseα (AMPKα) and its phosphorylated form Phosphorylated AMPKα Thr172 (p-AMPKα), Carnitine palmitoyl transferase 1 isoform A (CPT1A), Inhibitor of kappa B (IκB; 1/500) from Cell signalling; Peroxisome proliferator activated receptor gamma coactivator 1 alpha (PGC1α), Nuclear factor erythroid-2-related factor 2 (NRF2) from Abcam; Superoxide dismutase 2 (SOD2), Carbohydrate response element binding protein (ChREBP), glyceraldehyde 3-phosphate dehydrogenase (GAPDH) from Invitrogen; Glutathione peroxidase 1 (GPx), Cyclooxygenase-2 (COX-2) from Novus Biologicals; Nuclear factor kappa B (NF-κB; 1/250), Glucose 6-phosphatase (G6Pase), Phosphoenolpyruvate carboxykinase (PEPCK) from Santa Cruz Biotechnology; Phosphorylated acetyl-CoA carboxylase Ser79 (p-ACC; 1/500, Millipore); Tumor necrosis factor-alpha (TNFα; ThermoFisher Scientific, Waltham, MA, USA) and β-actin (1/250,000; Sigma-Aldrich, Burlington, MA, USA). Bands were captured and analyzed using a Chemidoc Imaging System coupled with an Image Lab software (Bio-Rad). For every protein of interest, β-actin or GAPDH was used for normalization of protein expression. Importantly, the results are most accurate when the forms of the protein of interest (total and phosphorylated) are applied to the same blot, and thus share the same β-actin. However, in order not to use stripping and re-probing procedure of the same membrane, which sometimes decreases the resolution, we loaded both forms of protein (total and phosphorylated for ACC and AMPKα) in different blots in some cases. Each of the forms was evaluated against its β-actin, before dividing the results of the phosphorylated form by the total form. Regarding the evaluation of NF-κB and IκB, the two proteins were developed using a same membrane with the housekeeping gene (GAPDH). The results thereby present each individual protein expression for comparison as well as the combined ratio.

### 2.7. Statistical Analysis

Data are expressed as mean ± SEM. For three-group comparison, statistical analysis was performed using one-way analysis of variance (ANOVA) with a post hoc Bonferroni multiple comparison test; for two-group comparison, Student *t*-Test was used. All statistical analyses were performed on IBM SPSS version 26.0 (IBM, Armonk, NY, USA, 2020). Results were considered statistically significant at *p* < 0.05.

## 3. Results

### 3.1. PACs and Obesity

To determine the effect of PACs on diet-induced obesity, male mice were fed with either a chow or a HFHS diet for 12 weeks. As noted in Figure 1A, from the beginning of feeding (week 1) to sacrifice (week 12), the HFHS-diet administration significantly increased body weight (152%) as compared to the chow group (121%). However, treatment with PACs lessened HFHS-mediated weight gain (140%, *p* < 0.001), independently of the total energy intake (Figure 1B). The same profile was observed in total adiposity and fat mass distribution (Figure 1C,D). However, no marked alterations were observed in gut (Figure 1F) and liver weight of HFHS-fed mice in response to PACs (Figure 1E). Thus, our results show that PACs exert an anti-obesity effect in a HFHS-treated murine model.

### 3.2. PACs and Glucose Dysmetabolism

We determined the influence of PACs on glucose homoeostasis and insulin sensitivity at week 8 in fasted animals. At the outset, we proceeded with the ipITT, designed to appraise the sensitivity of insulin responsiveness over time via blood glucose measurement, and following a bolus of intraperitoneal insulin. The HFHS group exhibited higher glucose levels during ipITT starting 5 min after insulin injection, reflecting decreased insulin sensitivity, whereas PACs-fed HFHS mice expressed a lower glycemic profile, indicating an improvement in insulin sensitivity (Figure 2A,B). The OGTT, performed at week 10, disclosed a similar glycemic pattern (Figure 2C,D) with a significant effect of PACs on insulinemia (Figure 2E,F) as well as HOMA-IR index measurement (Figure 2G). In fact, PACs strongly prevented HFHS-mediated IR as reflected by the decline of plasma insulin levels and HOMA-IR index, two effective indicators of glucose homoeostasis.

### 3.3. PACs and Lipid Metabolism

Mice fed a HFHS diet manifested dyslipidemia as reflected by higher lipid profile in plasma TG (Figure 3A) and TC (Figure 3B) as well as TG and TC in VLDL contents (Figure 3C,D) and non-HDL-cholesterol (Figure 3E). The administration of PACs alleviated HFHS-mediated dyslipidemia, with an elevated trend in levels of HDL-cholesterol (Figure 3F).

### 3.4. Impact of PACs on Circulatory Markers of Endotoxemia and Lipid Peroxidation

Since inflammation and OxS represent key factors in the induction of obesity and IR, we analyzed their basal systemic levels through their plasma biomarkers. As shown in Figure 4A, HFHS-diet contributed to raise the plasma LPS levels, a trigger of intestinal permeability and endotoxemia. While HFHS caused an accretion of LPS, PACs alleviated diet-induced LPS levels. We subsequently evaluated the status of F2-isoprostane and MDA (Figure 4B,C), two lipid peroxidation markers. Treatment with PACs normalized MDA and F2-isoprostane concentrations.

### 3.5. Effects of PACs on Lipid Accumulation in the Liver

A liver histologic examination of male mice under HFHS diet showed a vast number of fat droplets of different sizes, suggesting an increased lipid accumulation compared to chow group. However, PACs-treatment reduced lipid accumulation, thereby alleviating hepatic steatosis (Figure 5A,C). Lipid determinations confirmed these observations given that PACs reduced TG (Figure 5D) and TC (Figure 5E) in the liver of HFHS-treated mice.

### 3.6. PACs and Dysregulation of Hepatic Lipid Metabolism-Associated Proteins in HFHS-Treated Mice

Having established key physiological and metabolic consequences of a HFHS diet and in order to better reveal PACs-mediated alleviations, we directed our following investigations as a two-group comparison. As HFHS and PACs dynamically modified fat content in the liver, it appeared reasonable to assess hepatic lipid metabolism while focusing on the mechanisms of action. We first analyzed a key lipogenic enzyme involved in lipid synthesis, namely ACC which catalyzes the ATP-dependent carboxylation of acetyl coenzyme A to form malonyl-CoA. Although a decreased trend was observed, the results of ACC protein expression never yielded statistical significance. To give more strength to our results, we increased the number of animals from n = 4 to n = 8, and still the results did not show significant differences. Nevertheless, PACs promoted the phosphorylation level of ACC (p-ACC) protein, as reflected by the increased p-ACC/ACC ratio (Figure 6C), known to inhibit hepatic lipogenesis. Consistently, PACs reduced the protein expression of FAS (Figure 6D), the second central enzyme of lipogenesis by catalyzing the conversion of malonyl CoA to palmitate. As ACC and FAS are downstream target proteins of SREBP1c and ChREBP, we evaluated the protein mass of these powerful transcription factors and noted their downregulation in response to PACs treatment (Figure 6E and Figure 8C), respectively. Overall, these data emphasize the role of PACs in potently modulating lipid metabolism by reverting lipid accumulation in the liver and associated complications.

A second way to decrease excessive lipid levels in the liver may originate from the increment of mitochondrial fatty acid (FA) β-oxidation. To test this hypothesis, we measured the expression of CPT1A, the rate-limiting enzyme of FA β-oxidation. Treatment with PACs significantly enhanced CPT1A protein mass (Figure 6F) as well as PPARα and PGC1α, two transcription factors regulating mitochondria dysfunction and biogenesis (Figure 6F–H) in the liver of HFHS-fed mice. These findings support the beneficial effects of PACs in regulating mitochondrial biogenesis and function. Moreover, since AMPKα represents a key regulator of FA β-oxidation and lipogenesis, we investigated its phosphorylation level, which inactivates ACC and induces CPT1A. As illustrated in Figure 6I–K, the administration of PACs resulted in the increase of AMPKα phosphorylation (p-AMPKα) as shown with the p-AMPKα/AMPKα ratio, suggesting energy homoeostasis with an impact on lipogenesis and FA β-oxidation.

### 3.7. Impact of PACs on Inflammation and OxS

As the pathogenesis of obesity, MetS and MAFLD implicate inflammation and OxS, it was imperative to assess their liver biomarkers. Our experiments documented the capacity of PACs to lower the protein expressions of COX-2 and TNFα, as well as of NF-κB, a potent mediator of inflammatory responses (Figure 7), as confirmed by the low NF-κB/IκB ratio (Figure 7E).

We then turned to probe the effects of PACs on OxS, which allowed us to observe an increase in the protein mass of the hepatic antioxidant enzymes GPx and SOD2 (Figure 7F,G). We also confirmed a PACs-mediated rise in the levels of NRF2, the robust endogenous antioxidant regulator implicated in cellular redox homoeostasis (Figure 7H). Collectively, our data validate the potential of PACs to act as powerful repressors of inflammation and OxS.

### 3.8. PACs and Amplified Gluconeogenesis in the Liver of HFHS-Fed Mice

As impairment of hepatic glucose homoeostasis plays a crucial role in the pathogenesis of CVD, we examined the gluconeogenic pathway through its rate-limiting enzyme expressions. Our data clearly showed the ability of PACs to downregulate G6Pase (Figure 8A) and PEPCK (Figure 8B) protein loads, thereby contributing to limiting hepatic gluconeogenesis.

## 4. Discussion

Fruits and vegetables contain large phytochemicals with health-promoting potential. Among them, cranberries constitute a great source of polyphenols, notably PACs which are considered as promising bioactive molecules against chronic ailments. Since PACs stand out from the variety and high number of polyphenols because of their therapeutic impacts on chronic disorders [15,33], efforts have been deployed in the present investigation to examine their potential role in improving metabolic health and liver steatosis using a MetS mouse model. Our findings emphasize the effectiveness of PACs to protect against diet-induced obesity (via marked reduction of adiposity), IR (as documented by alleviation of insulinemia, OGTT, ipITT and HOMA-IR), and hyperlipidemia/hyperlipoproteinemia (pending the homeostasis of cholesterolemia, hypertriglyceridemia, VLDL-TG, VLDL-TC, non-HDL-cholesterol), the increase of which is known as a MetS manifestation. In addition, our findings highlighted the capacity of PACs to lower plasma markers of OxS and inflammation, regarded as promoters of IR. Moreover, beneficial effects of PACs were observed in fighting liver pathogenesis when considering the histologic and biochemical reduction of lipid deposition in association with the diminution of hepatic OxS and inflammation, downregulation of lipogenic enzymes and upregulation of fatty acid (FA) β-oxidation through regulation of powerful transcription factors. Clearly, PACs attenuated MetS features and improved liver pathogenesis.

Our findings demonstrated that HFHS-PACs-fed mice disclosed a reduction of weight gain and a curtailment of body fat expansion. This anti-obesogenic effect was unrelated to total energy intake, thereby excluding the involvement of central appetite pathways. Plausible mechanisms triggered by PACs could include (i) the binding of carbohydrates and proteins, thereby affecting their intestinal absorption [34,35]; (ii) inhibition of carbohydrate digestion and glucose absorption in the gut [36]; (iii) reduction in lipid emulsion and absorption in the gastrointestinal tract, consequently lowering calorie intake [37]; (iv) regulation of signalling pathways related to energy metabolism and adipogenesis [38,39]; (v) modifications of gut microbiota [40,41]; (vi) inhibition of the differentiation and proliferation of preadipocytes [42]; and (vii) induction of lipolysis and lipid metabolism [42]. Our results agree with an earlier report on the anti-obesogenic effects of cranberry PACs [28]. Further studies are obviously needed to reveal the precise mechanisms by which PACs produce adipose tissue remodeling and counteract obesity, thus providing further insight into the regulation of MetS development.

One of the purposes of our studies was to investigate the effects of PACs on IR, which represents a dominant trait of MetS [43] and a key process in the onset of type 2 diabetes [44,45]. Our observations pointed out that PACs attenuated intolerance to a glucose load via the prevention of IR as exhibited by our experiments. Probably, the mechanisms are represented by the protective effects of PACs-based treatment against OxS and inflammation, two interrelated factors impairing insulin action and mediating IR. As such, the beneficial impact of PACs on IR may originate from the downsizing of adipose tissue expansion that is able to disrupt the balance of its local by-products (i.e., adipokines) and consequently promote IR [46,47,48]. 

As revealed from our data, PACs display hypolipidemic effects as they relieved HFHS-mediated hypertriglyceridemia and hypercholesterolemia, and lessened mouse VLDL vehicles. Given that mice display low concentrations of low-density lipoprotein (LDL), we calculated non-HDL-cholesterol value, a more accurate and reliable depiction of atherosclerotic risk, as it encompasses all apolipoprotein-B containing lipoproteins, while representing a steadier parameter and predictor of LDL-cholesterol for hardening of the arteries [49,50]. Our experiments showed the effective potential of PACs to normalize HFHS-mediated elevated non-HDL-cholesterol concentrations. Although validation of our observations should be obtained in genetically modified mice (with LDL receptor or apolipoprotein E knockout, known to develop atherosclerosis), our findings clearly highlighted hyperlipidemia repression by PACs.

Our experiments provide evidence of the direct action of PACs at least on three MetS risk factors such as obesity, IR and dyslipidemia, as well as OxS and inflammation, two well-defined entities tied to MetS and CVD pathogenesis [51]. Since MAFLD has been redefined as a dysfunction associated with MetS [52,53], it was important to highlight the effects of PACs on MAFLD progression in our mouse model of dietary-induced MetS. Consistent with the improvement of ipITT (reflecting insulin sensitivity of the whole body, including liver), there were less HFHS-induced fat deposition and decreased TG and TC levels in response to PACs. Considering the relationship between liver steatosis and raised lipogenesis to the detriment of FA β-oxidation, we explored the role of PACs on these pathways by targeting their crucial regulating enzymes. Our observation was that PACs treatment led to lipogenic pathway alteration and β-oxidation activation. By which mechanisms were the expression of FA β-oxidation-related gene (CPT1A) enhanced, and the mass of lipogenic proteins (ACC and FAS) attenuated in the liver? According to our results, PACs downregulated the transcription factors, SREBP1c and ChREBP, necessary for the induction of the lipogenic genes and fatty liver development under hyperinsulinemia and a high-carbohydrate diet [54,55,56]. 

Interestingly, PACs were also able to activate AMPKα, a metabolic sensor of energy homoeostasis, which exerts simultaneous influence on lipogenesis and FA β-oxidation checkpoints. 

AMPKα activation by PACs likely resulted in ACC inhibition and CPT1A stimulation through their respective phosphorylation in the liver. The phosphorylated form of AMPKα (p-AMPKα) appears unchanged as described in the Results section. However, total form (AMPKα) is decreased following PAC supplementation, which leads to an increase in the p-AMPKα/AMPKα ratio in response to PACs. This significant increase shows that p-AMPKα is in advantageous quantity compared to total AMPKα molecules, which allows a higher stimulatory activity of AMPKα. It is usually the p-AMPKα/AMPKα ratio that is appreciated in the scientific literature.

To complete the protective mechanisms elicited by PACs for the prevention of liver fat deposition, we analyzed the protein masses of PPARα and PGC1α, two powerful proteins regulating mitochondrial biogenesis and function [57]. In fact, PACs enhanced PPARα, which is intrinsically involved in the uptake and oxidation of fatty acids via upregulation of CPT1A (critical for mitochondrial fatty acyl import). PACs also raised the PGC1α nuclear protein known as a transcriptional coactivator of PPARα and as a regulator for the stimulation of mitochondrial biogenesis and oxidative capacity. Therefore, it is reasonable to propose that PACs decrease the abundance of intrahepatic lipids by orchestrating the decline of de novo lipogenesis and increment of FA β-oxidation through the modulation of AMPKα/SREBP1c, ChREBP/ACC/FAS/CPT1A and PGC1α/PPARα/CPT1A signalling pathways.

Further investigations in the liver revealed that PACs thwarted PEPCK and G6Pase protein expression, thus limiting gluconeogenesis and glucose release. This effect may result from the modulation by ChREBP, a metabolic regulator that controls both lipid and glucose metabolism [58,59,60], mainly expressed in active sites of de novo lipogenesis such as liver and white and brown adipose tissues [61].

PACs treatment resulted in reduced protein expression of pro-inflammatory TNFα, known as a modulator of the immune system as it facilitates intercellular communication, proliferation, survival, differentiation, and apoptosis of leucocytes. Validation was obtained by decreased expression of COX-2, considered a crucial determinant of downstream mediators, which are detrimental to tissue integrity and contribute to liver damage [62]. The downregulation of these inflammatory agents by PACs is likely due to their action on the activation/suppression of NF-κB, the most prominent transcription factor of liver inflammation [63].

In response to PACs administration, there was a rise in endogenous GPx and SOD2, reflecting an improved antioxidant defense. PACs consistently upregulated NRF2, the central transcription factor that initiates the transcription of cytoprotective genes following binding to specific DNA sites termed antioxidant response elements [64]. In line with this finding, treatment with PACs led to enhanced protein expression of the transcriptional co-activator PGC1α that favors NRF2 induction, which could better promote endogenic hepatic defenses through antioxidative protection and mitochondrial biogenesis [65,66]. These data provide evidence of the protective effect of PACs via regulation of PGC1α–NRF2.

The daily dose of 200 mg/kg in mice would translate to 16 mg/kg/day in humans [67], corresponding to 1120 mg/day of PACs for a person weighing 70 kg. To achieve this intake, one would have to consume roughly 350 g of cranberries [68]. However, it must be stressed that the consumption of whole cranberries implies an amalgam of several polyphenols, while the aim of this paper focuses on examining the impact of a purified PAC supplementation. In this context, the present work represents a very good contribution to the advancement of knowledge on the action of the PAC class.

Collectively, the findings of this research exhibit promising efficiency of PACs in alleviating diet-induced MetS conditions, which have become highly severe health problems. PACs reversed weight gain by curtailing adipose tissues in different sites, ameliorated inflammation- and oxidative stress-induced insulin resistance while maintaining glucose homoeostasis and relieving hyperlipidemia, a fundamental risk factor for cardiovascular diseases. Furthermore, modulation by PACs of transcription factors governing metabolic pathways related to lipid metabolism, notably lipogenesis and β-oxidation, significantly suppressed hepatic lipid deposition and attenuated HFHS-mediated MAFLD. Therefore, our data provide support for the role of PACs as functional food in the battle against MetS and MAFLD. These results should be approached with caution, however, as the mechanisms highlighted in our experimental model represent probably just a segment of the metabolic disruptions associated with both MetS and MAFLD. Furthermore, whether PACs can display the same metabolic impact in humans clearly requires thorough clinical investigations.

## 5. Conclusions

Prevention and management of both MetS and MAFLD are currently costly and ineffective as the prevalence of these clusters continue to soar. Functional foods such as PACs could confer many advantages as multi-targeted agents in these disorders. In mice, PACs supplementation favors hepatic homoeostasis and attenuation of metabolic disturbances defining the MetS and CVD.

## Figures and Tables

**Figure 1 antioxidants-12-00090-f001:**
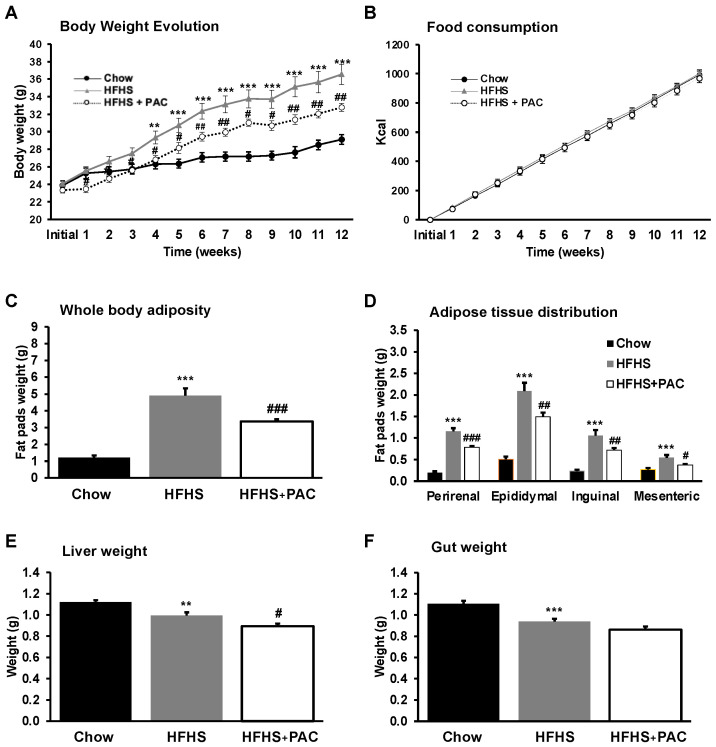
Effect of PAC on body weight gain, food intake and organs weight. Mice were fed either a standard chow diet or a high-fat, high-sucrose diet (HFHS) ± 200 mg/kg polyphenol proanthocyanidin-rich fraction (PAC) body weight/day by gavage (HFHS + PAC) for 12 weeks. Chow diet and HFHS-fed mice were gavaged with a water vehicle. (**A**) body weight gain, (**B**) energy intake, (**C**) whole adipose tissue weight (as a total of collected fat pads) and (**D**) its tissue distribution, (**E**) liver and (**F**) gut weights were measured as described in Materials and Methods. Results are presented as mean ± SEM for n = 12–16 mice/group. ** *p* < 0.01, *** *p* < 0.001 vs. chows; # *p* < 0.05, ## *p* < 0.01, ### *p* < 0.001 vs. HFHS mice.

**Figure 2 antioxidants-12-00090-f002:**
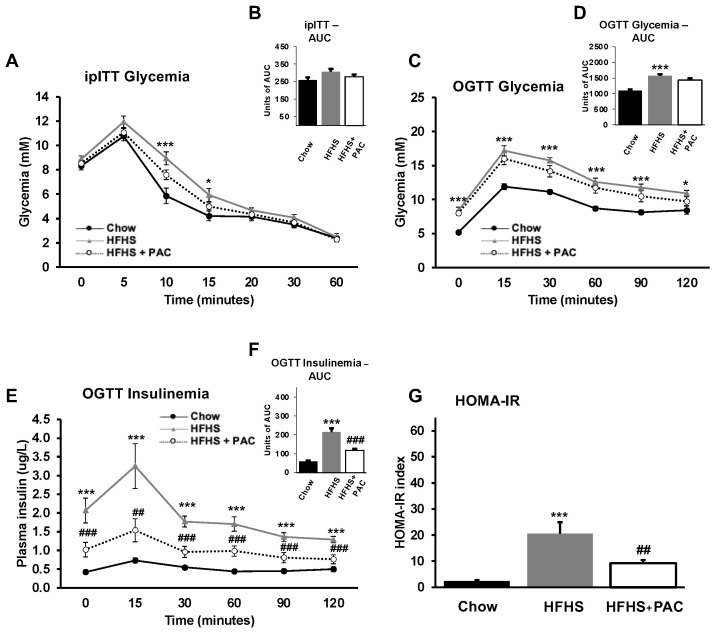
Effect of PAC on glucose homoeostasis and insulin resistance. After 8-week diet, mice were submitted to an intra-peritoneal (**A**,**B**) insulin tolerance test (ipITT) following a 6 h fast. At week 10, mice were then submitted to an (**C**,**D**) oral glucose tolerance test (OGTT) following an overnight fast to evaluate (**E**,**F**) insulinemia. (**G**) The Homeostatic model assessment of insulin resistance (HOMA-IR) was calculated as described in Materials and Methods. Results are presented as mean ± SEM for n = 12–16 mice/group. * *p* < 0.05, *** *p* < 0.001 vs. chows; ## *p* < 0.01, ### *p* < 0.001 vs. HFHS mice.

**Figure 3 antioxidants-12-00090-f003:**
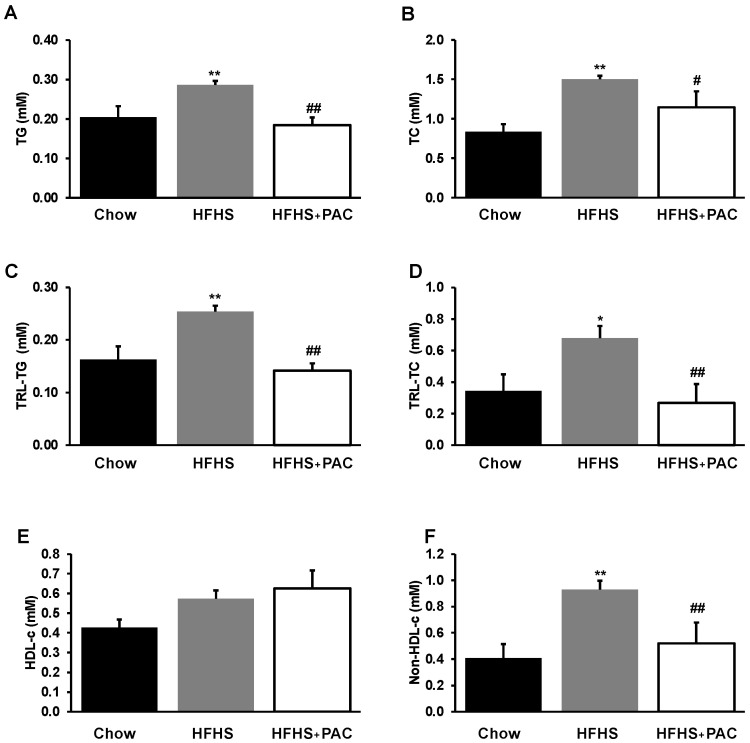
Effect of PAC on dyslipidemia and lipoprotein composition. After 12 weeks diet and before sacrifice, plasma was collected for lipid profile determination. (**A**) triglycerides (TG), (**B**) total cholesterol (TC) as well as (**C**) TG, (**D**) TC in triglyceride-rich lipoprotein fractions (TRL) content, (**E**) HDL-cholesterol and (**F**) non-HDL-cholesterol were analyzed as described in Materials and Methods. Results are presented as mean ± SEM for n = 4–6 pooled plasma/group. * *p* < 0.05, ** *p* < 0.01 vs. chows; # *p* < 0.05, ## *p* < 0.01 vs. HFHS mice.

**Figure 4 antioxidants-12-00090-f004:**
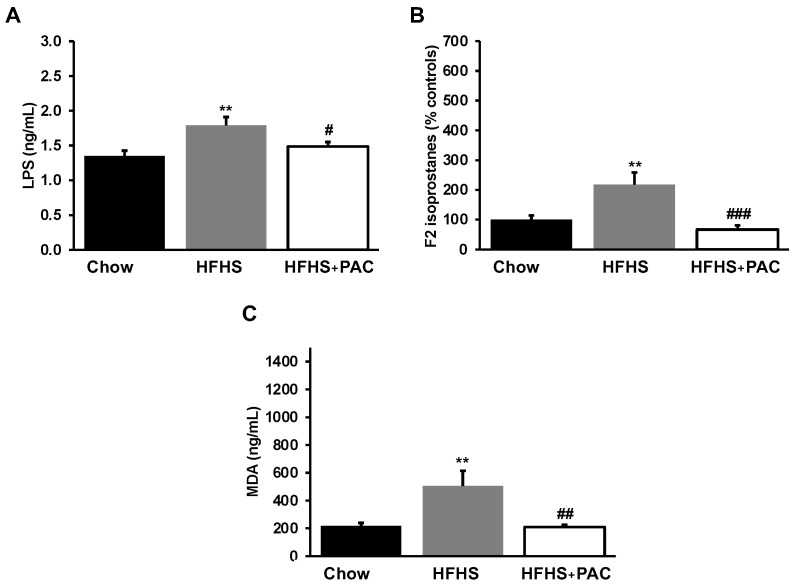
Impact of PAC in inflammatory and OxS markers in HFHS-fed mice. At sacrifice, plasma (**A**) lipopolysaccharide (LPS), (**B**) F2-isoprostanes and (**C**) malondialdehyde (MDA) concentrations were determined as described in Materials and Methods. Results are presented as mean ± SEM for n = 12–16 mice/group. ** *p* < 0.01 vs. chows; # *p* < 0.05, ## *p* < 0.01, ### *p* < 0.001 vs. HFHS mice.

**Figure 5 antioxidants-12-00090-f005:**
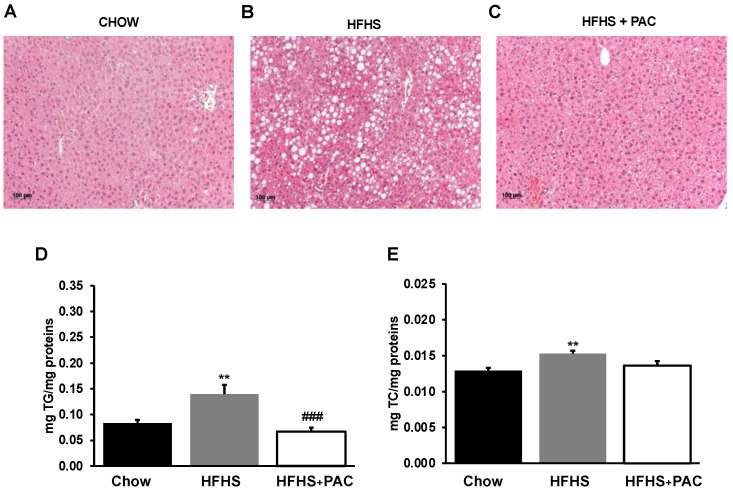
Effects of PAC on hepatic lipid accumulation. Representative images of hematoxylin phloxine saffron stained liver sections of (**A**) chow-, (**B**) high-fat, high-sucrose (HFHS) and (**C**) HFHS + PAC-fed mice for n = 4 mice/group; (**D**) Triglycerides and (**E**) total cholesterol content was quantified in liver tissue for n = 8 mice/group. Results are presented as mean ± SEM. ** *p* < 0.01 vs. chows; ### *p* < 0.001 vs. HFHS mice.

**Figure 6 antioxidants-12-00090-f006:**
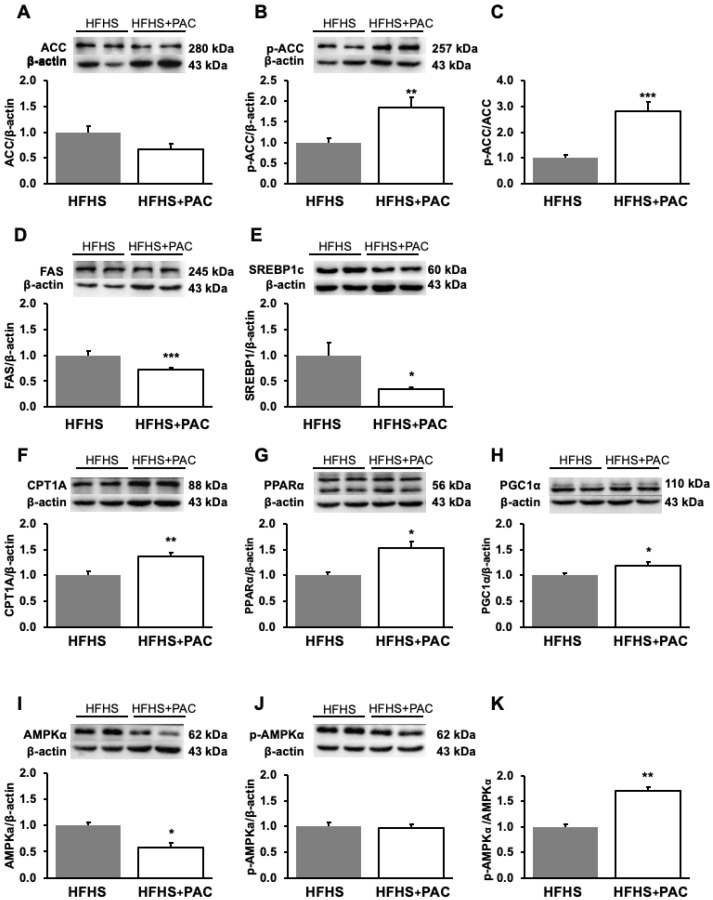
PAC modulates hepatic lipid metabolism. Protein expression of pivotal biomarkers influencing lipid metabolism and biogenesis was determined in liver by Western blot as described in Materials and Methods. (**A**) ACC, (**B**) p-ACC, (**D**) FAS, (**E**) SREBP1c, (**F**) CPT1A, (**G**) PPARα, (**H**) PGC1α, (**I**) AMPKα and (**J**) p-AMPKα protein mass was determined. (**A**) ACC and (**B**) p-ACC as well as (**I**) AMPKα and (**J**) p-AMPKα were blotted on separate gels and normalized using their respective β-actins; the (**C**) p-ACC/ACC and (**K**) p-AMPKα/AMPKα ratios were then calculated. Data are expressed as the mean ± SEM for n = 4–8 mice/group with a representative gel per experiment. * *p* < 0.05, ** *p* < 0.01, *** *p* < 0.001 vs. HFHS mice. Each set of experiments, including the proteins of interest and their β-actin (as a reference protein and a loading housekeeping control) was run on the same gel. ACC: acetyl-CoA carboxylase; p-ACC: Phosphorylated-acetyl-CoA carboxylase; FAS: Fatty acid synthase; SREBP1c: Sterol regulatory element binding protein-1c; CPT1A: Carnitine palmitoyl transferase 1 isoform A; PPARα: Peroxisome proliferator activated receptor alpha; PGC1α: Peroxisome proliferator activated receptor gamma coactivator 1 alpha; AMPKα: AMP-activated protein kinaseα; p-AMPKα: Phosphorylated-AMP-activated protein kinaseα.

**Figure 7 antioxidants-12-00090-f007:**
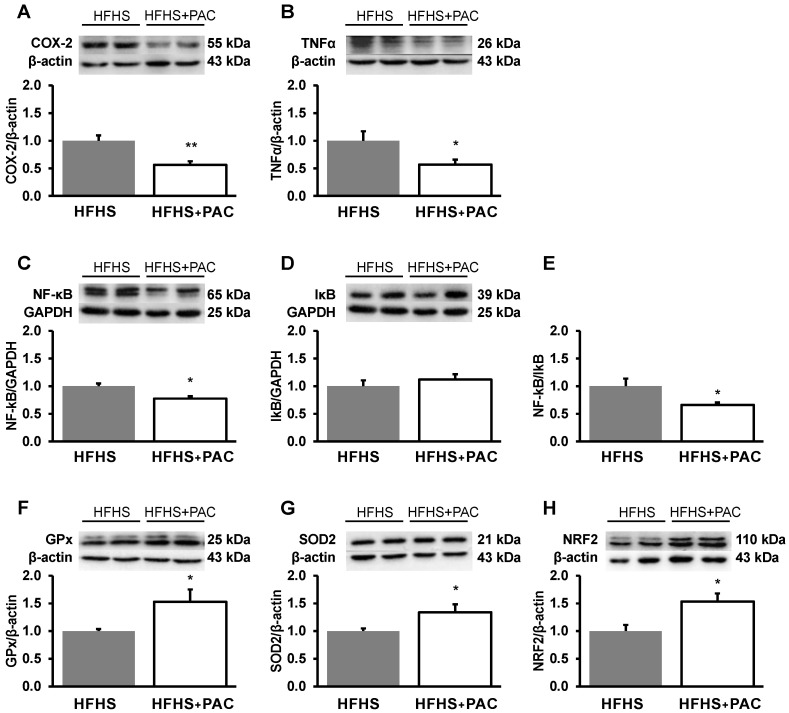
Effect of PACs on inflammatory and OxS processes in liver HFHS-fed mice. The protein mass of (**A**) COX-2, (**B**) TNFα, (**C**) NF-κB and (**D**) IκB was determined by Western blot as described in Materials and Methods. Both (**C**) NF-κB and (**D**) IκB were blotted on the same gel and normalized using the same GAPDH; the (**E**) NF-kB/IκB ratio was then calculated. Moreover, the protein expression of the antioxidant defense biomarkers, namely (**F**) GPx, (**G**) SOD2 and the transcription factor nuclear factor (**H**) NRF2 were evaluated by Western blot as described in Materials and Methods section. Results are presented as mean ± SEM for n = 4–8 mice/group with a representative gel per experiment. * *p* < 0.05, ** *p* < 0.01 vs. HFHS mice. Each set of experiments, including the proteins of interest and their β-actin or GAPDH (as a reference protein and a loading housekeeping control) was run on the same gel.

**Figure 8 antioxidants-12-00090-f008:**
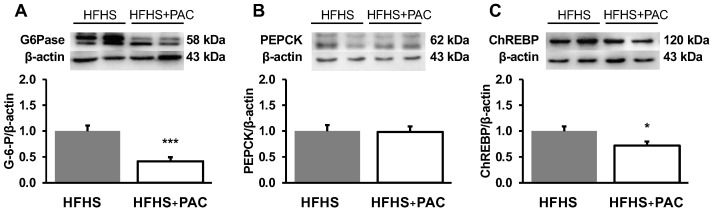
PACs modulate glucose metabolism in liver HFHS-fed mice. Protein mass of (**A**) G6Pase, (**B**) PEPCK as well as the transcriptional factor (**C**) ChREBP was assessed by Western blot as described in Materials and Methods. Data are expressed as the mean ± SEM for n = 4–7 mice/group with a representative gel per experiment. * *p* < 0.05, *** *p* < 0.001 vs. HFHS mice. Each set of experiments, including the proteins of interest and their β-actin (as a reference protein and a loading housekeeping control) was run on the same gel. G6Pase: Glucose 6-phosphatase; PEPCK: Phosphoenolpyruvate carboxykinase; ChREBP: Carbohydrate response element binding protein.

## Data Availability

All of the data is contained within the article and the Appendix A.

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
