# Peer review of "Cranberry Proanthocyanidins as a Therapeutic Strategy to Curb Metabolic Syndrome and Fatty Liver-Associated Disorders"

_antioxidants, 2022, doi:10.3390/antiox12010090_

Round 1

Reviewer 1 Report

This is an interesting study evaluating the beneficial effect of cranberry  proathocyanidins on metabolic syndrome and MAFLD. This mauscript is well written and experiments are conducted properly. 

1. I wonder if NF-kappa B molecular weight is correct. the main component is p65 and p50, 65 and 50 kilodaltons, respectively. Please check it. 

2. There are some typographic errors in the text. Please correct them before resubmission. 

Author Response

This is an interesting study evaluating the beneficial effect of cranberry proanthocyanidins on metabolic syndrome and MAFLD. This manuscript is well written and experiments are conducted properly. 

We appreciate the Reviewer's words of encouragement.

  1. I wonder if NF-kappa B molecular weight is correct. The main component is p65 and p50, 65 and 50 kilodaltons, respectively. Please check it. 

We thank the Reviewer for his vigilance. Corrections were made to identify the true molecular weight of p65.

  1. There are some typographic errors in the text. Please correct them before resubmission. 

The text has been revised and typographical errors have been corrected. 

Reviewer 2 Report

The paper by Feldman and coworkers is an investigation in the use of cranberry extract as a dietary supplement to alleviate MAFLD.  The authors demonstrate that cranberry proanthrocyanidins (PAC) attenuate some of the physical maladies of a high fat high sucrose diet in terms of weight gain and fat accumulation.  PACs also affect some of the signaling pathways involved with fatty acid synthesis, beta-oxidation, inflammation, etc.  Most of the data in this paper are noteworthy and credible.  However, there are a few weaknesses that should be addressed before the paper is ready for publication.  In particular, some of the WBs in the paper need to be verified and may need to be redone as the methodology of precisely how they were handled is not clear.  Overall, this is a very interesting study and more so if more people liked the taste of cranberries.

·         Methodology for whole body adiposity and adipose tissue distribution is not described.  There is a brief mention of liver and gut weight so I don’t know if these were calculated in the adiposity data.

·         Fig. 5:  The instrumentation for image capture was mentioned but not the software or method used to assess the tissue.

·         Fig. 6A:  I would suspect that if the desitometry of the load control, b-actin, were normalized, the bands for ACC  HFHS+PAC would be much lower and significantly different.

·         Fig. 6C & 6K:  Where are the Western blots?  It is typical to show phosphor-X and total-X and the load control on one blot in which the blot is first probed for the phosphorylated protein and then stripped and re-probed for total protein.  If this method was performed, it should be stated.  If the ratio is just calculated from 2 different blots, that is not very accurate and should the Westerns should be re-done.  For reduction of samples, the use of DTT followed by iodoacetamide treatment to prevent reoxidation of the disulfide bonds is better for Westerns than the use of B-Me.

·         Fig. 7C:  Blot should be redone. 

·         Fig. 7F:  See response to 6C and 6K

·         Discussion: since AMPK is so important in the signaling pathways for which the hypothesis of the paper relies on, I am not entirely convinced that AMPK signaling is increased in the HFHS+PAC samples due to the Western blots in Fig. 6.

·         The mice were fed a powdered extract of 200 mg/kg.  What would that translate into human consumption of the weight of cranberries that a 70 kg person would need to consume on a daily basis?

·         There are numerous papers touting an extract of some berry, or Chinese herb or other natural compound such as grape seed that alleviates fatty liver, or NASH , or NAFLD in some of the same signaling pathways as described in this paper.  How is this experimental study similar or different from the others?  This can be included in the discussion.

·         References are suitable, however, #3 should be updated as Z.M. Younossi has more current papers on the subject.

Author Response

Most of the data in this paper are noteworthy and credible.  However, there are a few weaknesses that should be addressed before the paper is ready for publication.  In particular, some of the WBs in the paper need to be verified and may need to be redone as the methodology of precisely how they were handled is not clear.  Overall, this is a very interesting study and more so if more people liked the taste of cranberries.

  • We thank the Reviewer for his constructive comments. We have of course taken into consideration his recommendations and have redone the Western blots of Figure 7C (IL-6). The results of the Western blots did not improve even if we had changed the batch of antibodies. Therefore, we have removed this protein from our results and have adjusted the manuscript accordingly. For each of the other elements, we have provided the explanations.

Methodology for whole body adiposity and adipose tissue distribution is not described.  There is a brief mention of liver and gut weight so I don’t know if these were calculated in the adiposity data.

There is a real distinction between measurements of adipose tissue and other organs such as the liver and intestine. Each region of adipose tissues was separately collected and weighed, allowing the distribution of the overall adipose tissue to be established. Subsequently, all the weights of the individual fat tissues were added together to determine the total amount of adipose tissue. Importantly, the liver and gut were treated separately. This information is clearly described in the Methods section.

Fig. 5:  The instrumentation for image capture was mentioned but not the software or method used to assess the tissue.

For the tissue histology, the liver specimens were fixed in 10% formalin overnight and then embedded in paraffin. Sections (3 µm) were obtained with a microtome and immediately stained with hematoxylin-eosin, and examined by light microscopy. Images of the stained tissues were captured with a Zeiss Imager A1 and measurements were evaluated with the Axiovision software. This information has now been included in the Methods section.

  • Fig. 6A:  I would suspect that if the densitometry of the load control, b-actin, were normalized, the bands for ACC, HFHS+PAC would be much lower and significantly different.

Il is important to note that all bands the proteins of interest were normalized as a function of their respective b-actin. Although a trend was observed, the results of ACC protein expression have never shown statistical significance. To give more strength to our results, we increased the number of animals from n=4 to n=8, and still the results did not show significant differences.

Fig. 6C & 6K:  Where are the Western blots?  It is typical to show phosphor-X and total-X and the load control on one blot in which the blot is first probed for the phosphorylated protein and then stripped and re-probed for total protein.  If this method was performed, it should be stated.  If the ratio is just calculated from 2 different blots, that is not very accurate and should the Westerns should be re-done.  For reduction of samples, the use of DTT followed by iodoacetamide treatment to prevent reoxidation of the disulfide bonds is better for Westerns than the use of B-Me.

The Reviewer is right in mentioning that the results are most accurate when the forms of the protein of interest (total and phosphorylated) are applied to the same blot, and thus share the same beta-actin.  However, in order not to use stripping and reprobing procedure of the same membrane, which sometimes decreases the resolution, we loaded both forms of protein (total and phosphorylated) in different blots. Each of the forms was evaluated against its beta-actin, before dividing the results of the phosphorylated form by the total form. This approach is also valid and is often observed in the vast scientific literature.

Fig. 7C:  Blot should be redone. 

We have repeatedly tried to reveal IL-6 with our antibody without getting satisfactory results. Following the Reviewer's comments, we even changed the antibody, but the IL-6 protein remained blurred and of poor quality. We therefore decided to discard this figure to avoid ambiguity,

Fig. 7F:  See response to 6C and 6K

Regarding the evaluation of NF-kB and IkB, the two proteins were developed using a single blot (unlike the ACC and AMPK).  Therefore, both proteins (NF-kB and IkB) share the same reference protein (GAPDH). This information has been made clearer in the manuscript and in the supplementary materials.

  • Discussion: since AMPK is so important in the signaling pathways for which the hypothesis of the paper relies on, I am not entirely convinced that AMPK signaling is increased in the HFHS+PAC samples due to the Western blots in Fig. 6 (page 13).

We agree with the Reviewer that the phosphorylated form of AMPK (p-AMPK) appears unchanged as described in the Results section. However, total form (AMPK) is decreased following PAC supplementation, which leads to an increase in the p-AMPK/AMPK ratio in response to PACs. This significant increase shows that p-AMPK is in advantageous quantity compared to total AMPK molecules, which allows a higher stimulatory activity of AMPK. It is usually the pAMPK/AMPK ratio that is appreciated in the scientific literature (page 13).

  • The mice were fed a powdered extract of 200 mg/kg.  What would that translate into human consumption of the weight of cranberries that a 70 kg person would need to consume on a daily basis? There are numerous papers touting an extract of some berry, or Chinese herb or other natural compound such as grape seed that alleviates fatty liver, or NASH , or NAFLD in some of the same signaling pathways as described in this paper.  How is this experimental study similar or different from the others?  This can be included in the discussion.

A daily dose of 200mg/kg in mice would translate to 16mg/kg/day in humans [ Basic Clin Pharm 2016, 7, 27-31], corresponding to 1120mg/day of PACs for a person weighting 70kg. To achieve this intake, one would have to consume roughly 350g of cranberries [ USDA Database for the Proanthocyanidin Content of Selected Foods. 2018]. However, it must be stressed that consumption of whole cranberries implies an amalgam of several polyphenols, while the aim of this paper focuses on examining the impact of a purified PAC supplementation. In this Context, in this context, the present work represents a very good contribution to the advancement of knowledge on the action of this class of polyphenols (PACs) (page 14).

       References are suitable, however, #3 should be updated as Z.M. Younossi has more current papers on the subject.

      The more recent paper headed by Younossi has been incorporated.

Round 2

Reviewer 2 Report

My concerns were addressed by the authors

Author Response

  1. I wonder if NF-kappa B molecular weight is correct. the main component is p65 and p50, 65 and 50 kilodaltons, respectively. Please check it. 

The reviewer is absolutely correct in stating that the main component of p65 is 65kDa (75 kDa rather referring to the loading protein used to identify the gel). The image has been corrected.

  1. There are some typographic errors in the text. Please correct them before resubmission. 

            Typographic, spelling and grammar errors have been reviewed.

Line 28: Typographic mistake corrected

Line 30: Spelling mistake corrected

Line 43: Typographic mistake corrected

Line 51: Grammar mistake corrected

Line 75: Spelling mistake corrected

Line 92: Grammar mistake corrected

Line 93:Typographic mistake corrected

Line 106: Spelling mistake corrected

Lines 108-113:Typographic mistake corrected

Lines 143-152: Typographic mistake corrected

Line 201:Typographic mistake corrected

Line 202:Typographic mistake corrected

Line 210:Typographic mistake corrected

Line 212: Spelling mistake corrected

Line 237: Grammar mistake corrected

Line 248 : Figure 1: typographic mistakes corrected in the figure itself

Line 252:Typographic mistake corrected

Line 269: Spelling mistake corrected

Line 275: Grammar mistake corrected

Line 276: Grammar mistake corrected

Line 294:Typographic mistake corrected

Line 295: Spelling mistake corrected

Line 296: Spelling mistake corrected

Line 302: Spelling mistake corrected

Line 324: Grammar mistake corrected

Line 342: Spelling mistake corrected

Line 348: Typographic mistakes corrected

Line 353: Grammar mistake corrected

Line 372: Typographic mistake corrected

Line 373: Spelling mistake corrected

Line 378-381: Typographic mistakes corrected

Line 409: Figure 7C: molecular weight of NF-kB modified to 65kDa

Line 485: Spelling mistake corrected

Line 500: Spelling mistake corrected

Line 512: Spelling mistake corrected

Line 558: Typographic mistake corrected

Line 595: Typographic mistake corrected

Line 598: Typographic mistake corrected

Line 599: Spelling mistake corrected

Line 603: Grammar mistake corrected

Line 672: Typographic mistake corrected
